# Shifts in methanogenic archaea communities and methane dynamics along a subtropical estuarine land use gradient

Sebastian Euler[1]*, Luke C. Jeffrey[1], Damien T. Maher[1,2], Derek Mackenzie[1], Douglas R. Tait[1]

**1** SCU GeoScience, Southern Cross University, Lismore, NSW, Australia, **2** School of Environment, Science and Engineering, Southern Cross University, Lismore, NSW, Australia

\* euler.sebastian@gmail.com, s.euler.10@student.scu.edu.au

**Data Availability Statement:** Environmental and microbial data used in the plots presented are available in the attached files (.csv format). Raw sequencing data are available from the Knowledge

## Abstract

In coastal aquatic ecosystems, prokaryotic communities play an important role in regulating the cycling of nutrients and greenhouse gases. In the coastal zone, estuaries are complex and delicately balanced systems containing a multitude of specific ecological niches for resident microbes. Anthropogenic influences (i.e. urban, industrial and agricultural land uses) along the estuarine continuum can invoke physical and biochemical changes that impact these niches. In this study, we investigate the relative abundance of methanogenic archaea and other prokaryotic communities, distributed along a land use gradient in the subtropical Burnett River Estuary, situated within the Great Barrier Reef catchment, Australia. Microbiological assemblages were compared to physicochemical, nutrient and greenhouse gas distributions in both pore and surface water. Pore water samples from within the most urbanised site showed a high relative abundance of methanogenic *Euryarchaeota* (7.8% of all detected prokaryotes), which coincided with elevated methane concentrations in the water column, ranging from 0.51 to 0.68 µM at the urban and sewage treatment plant (STP) sites, respectively. These sites also featured elevated dissolved organic carbon (DOC) concentrations (0.66 to 1.16 mM), potentially fuelling methanogenesis. At the upstream freshwater site, both methane and DOC concentrations were considerably higher (2.68 µM and 1.8 mM respectively) than at the estuarine sites (0.02 to 0.66 µM and 0.39 to 1.16 mM respectively) and corresponded to the highest relative abundance of methanotrophic bacteria. The proportion of sulfate reducing bacteria in the prokaryotic community was elevated within the urban and STP sites (relative abundances of 8.0%– 10.5%), consistent with electron acceptors with higher redox potentials (e.g. $O_2$, $NO_3^-$) being scarce. Overall, this study showed that ecological niches in anthropogenically altered environments appear to give an advantage to specialized prokaryotes invoking a potential change in the thermodynamic landscape of the ecosystem and in turn facilitating the generation of methane–a potent greenhouse gas.

Network for Biocomplexity (doi:10.5063/F1SQ8XSB).

**Funding:** SE RGRG1900022 Great Barrier Reef Marine Park Authority gbrmpa.gov.au The funders had no role in study design, data collection and analysis, decision to publish, or preparation of the manuscript. DT DE180100535 Australian Research Council arc.gov.au The funders had no role in study design, data collection and analysis, decision to publish, or preparation of the manuscript.

**Competing interests:** The authors have declared that no competing interests exist.

# 1 Introduction

Land use is rapidly changing coastal environments with estuaries now representing one of the most altered and vulnerable ecosystems on the planet [1]. Estuaries are biogeochemical hotspots for carbon and nutrient cycling with nutrient inputs from a range of sources including terrestrial, riverine and groundwater [2, 3]. These nutrients are predominantly processed by bacteria and archaea within the sediments and water column of ecosystems, which can release atmospheric greenhouse gases (GHG's) such as carbon dioxide ($CO_2$) and methane ($CH_4$) [4–6]. High loading rates of organic carbon and other nutrients in areas affected by urban or agricultural land use can potentially increase emission of GHG's, compared to their pristine counterparts [7, 8].

Estuaries contribute between 1 and 7 Tg of $CH_4$ and 0.1 to 0.25 Gt of $CO_2$ to the atmosphere each year [9], with the global flux of $CO_2$ to the atmosphere from estuaries comparable to the uptake of the entire continental shelf, despite estuaries representing only 5% of the continental shelf equivalent surface area [10]. Increasing inputs of anthropogenic pollutants stemming from urban, industrial and agricultural runoff into adjacent estuarine ecosystems have been reported to elevate GHG fluxes [11–13]. $CH_4$ emissions originating from microbial sources have been suggested to contribute about 70% of all global methane emissions [14–16]. Due to the dynamic nature and spatial heterogeneity of estuarine GHG's, the underlying drivers, mechanisms and direct comparisons to microbiomes remain poorly understood [17].

Prokaryotic communities in estuaries can be complex and primarily consist of unculturable lineages, which makes laboratory-based research challenging [6, 18, 19]. Natural and anthropogenic gradients in estuarine ecosystems can be an ideal environment for gauging the response of microbes to environmental variability [20, 21]. Links between microbial communities and basic physicochemical parameters such as salinity and dissolved oxygen (DO) have been previously demonstrated in estuarine environments [21–23]. For example, Hong et al. [22] studied microbial communities in a subterranean estuary of Gloucester Beach, United States and found that the shift in microbial community compositions was mainly driven by variations in physicochemical parameters (DO, salinity, temperature). However, comprehensive investigations to further our understanding of microbial ecology and how it drives nutrient and GHG cycling under land use change are required [21, 24].

Archaeal diversity and abundance is underexplored in coastal ecosystems and plays an important role in the dynamics of GHG's and especially $CH_4$ production [25–27]. Methanogenic archaea are able to convert $H_2$, acetate, $CO_2$ and other carbon compounds (e.g. CO, formate, methanol, methylamines) into $CH_4$ [25, 28, 29]. Dissolved organic carbon (DOC) can be broken down microbially via hydrolysis, acidogenesis and acetogenesis prior to methanogenesis [30, 31]. Thermodynamically, methanogenesis is associated with a small free energy change, allowing for the synthesis between 1 (for acetoclastic methanogenesis) and 2 ATP under standard conditions, and less than 1 ATP under most environmental conditions [14, 29]. Methanogens inhabit a unique ecological niche and are highly adapted for thermodynamic energy conservation [29, 32]. Within the estuarine continuum, they likely reside and produce $CH_4$ in impacted locations characterized by high loads of DOC and a thermodynamic landscape that is unfavourable to many other microbes.

To gauge the factors contributing to the proliferation of specific prokaryotes occupying estuarine ecological niches, the competition between different functional groups plays an important role [33, 34]. For example, methanogens are readily outcompeted by microbial taxa utilizing more thermodynamically favourable electron acceptors such as oxygen, nitrate, iron and sulfate [35–37]. Denitrification has been shown to diminish methanogenesis via denitrifiers directly outcompeting methanogenic archaea [35, 38, 39]. Sulfate reducing bacteria do not always impact methanogen abundance and may only outcompete the latter for $H_2$ and acetate,

but not for labile DOC compounds like methylamines [39–41]. Recent studies have shown syntrophic interactions between methanogens and sulfate reducers co-existing in the same environment [34, 42]. This cross-feeding, however, can be accompanied by elevated GHG emissions from heavily modified coastal ecosystems [5].

A proportion of the $CH_4$ produced through methanogenesis can be offset by methanotrophic bacteria. Methanotrophs are divided into phylogenetically distinct types. The proteobacterial type I (*Methylococcales* order) and type II (*Methylocystaceae* family) methanotrophs are both able to use $CH_4$ as their sole carbon and energy source [43, 44]. Additionally, new kinds of acidophilic methanotrophs have recently been discovered within the *Verrucomicrobia* phylum (the *Methylacidiphilales* order) [45]. When not oxidized by methanotrophs, $CH_4$ can escape into the atmosphere [43]. Therefore, understanding the balance between methanogenesis and methanotrophy is important in constraining $CH_4$ emissions from impacted coastal ecosystems.

In this study we investigate the phylogenetic prevalence of $CH_4$ producing archaea relative to other relevant prokaryotic communities within the surface and pore water of a subtropical estuarine land use gradient (Burnett River, Australia). We correlate these microbiological assemblages with in-depth biogeochemical characterisations including physicochemical parameters (DO, salinity, temperature), nutrient availability (DOC, ammonium, nitrate and sulfate) and the GHG's $CH_4$ and $CO_2$. This multi-parameter and multi-disciplinary approach is then used to determine the influence of different land uses along the estuarine continuum on microbial and abiotic factors.

## 2 Materials and methods

### 2.1 Study site

The Burnett River estuary, situated in the subtropical Great Barrier Reef (GBR) catchment area, features a multitude of different ecological and land use zones along its length, before discharging into the Coral Sea on Australia's east coast (Fig 1). Situated adjacent to the Great Dividing Range, the region features an elevated topography with floodplain areas adjacent to the river (e.g. around Bundaberg). The subtropical climate receives a mean annual rainfall of ~1000 mm (www.bom.gov.au) with approximately one third of the rainfall occurring in summer between January and February. The average annual temperature in the region is 21.5°C with a mean maximum of 26.8°C and a mean minimum of 16.4°C. Anthropogenic development in the Burnett River estuary catchment includes widespread agricultural land use including livestock grazing and horticulture (predominantly sugarcane), as well as mining and urban development [46, 47]. A total of ~2,800,000 ha or 74% of the Burnett Catchment area is used by agriculture, with grazing accounting for ~80% of agricultural land use, while urban development (mainly the city of Bundaberg) accounts for ~13%.

The estuary mouth (Site 1) is mostly pristine, with the Barubbra Island Regional Park spanning the entire northern shore and low-density dwellings situated on the southern bank (Fig 1). Upstream from the mouth, land use is predominantly agricultural with livestock grazing and horticulture on both sides of the river (Site 2). About 13 km upstream (Site 3), the city of Bundaberg supports a population of ~70,000 (or 232 people per $km^2$) with significant urban and industrial development (predominantly sugar industry related, e.g. distilling and cane harvester manufacturing) around the estuary. Mangroves line parts of the lower estuary channel which is bordered by agricultural and urban land uses (Sites 2, 3 & 4). Situated at the upper city boundary (Site 4), a sewage treatment plant (STP) discharges treated wastewater into the main channel. Approximately 25 km from the ocean, a concrete weir (Site 5—downstream and Site 6—upstream) moderates the tidal limit of the estuary. The weir causes a steep salinity

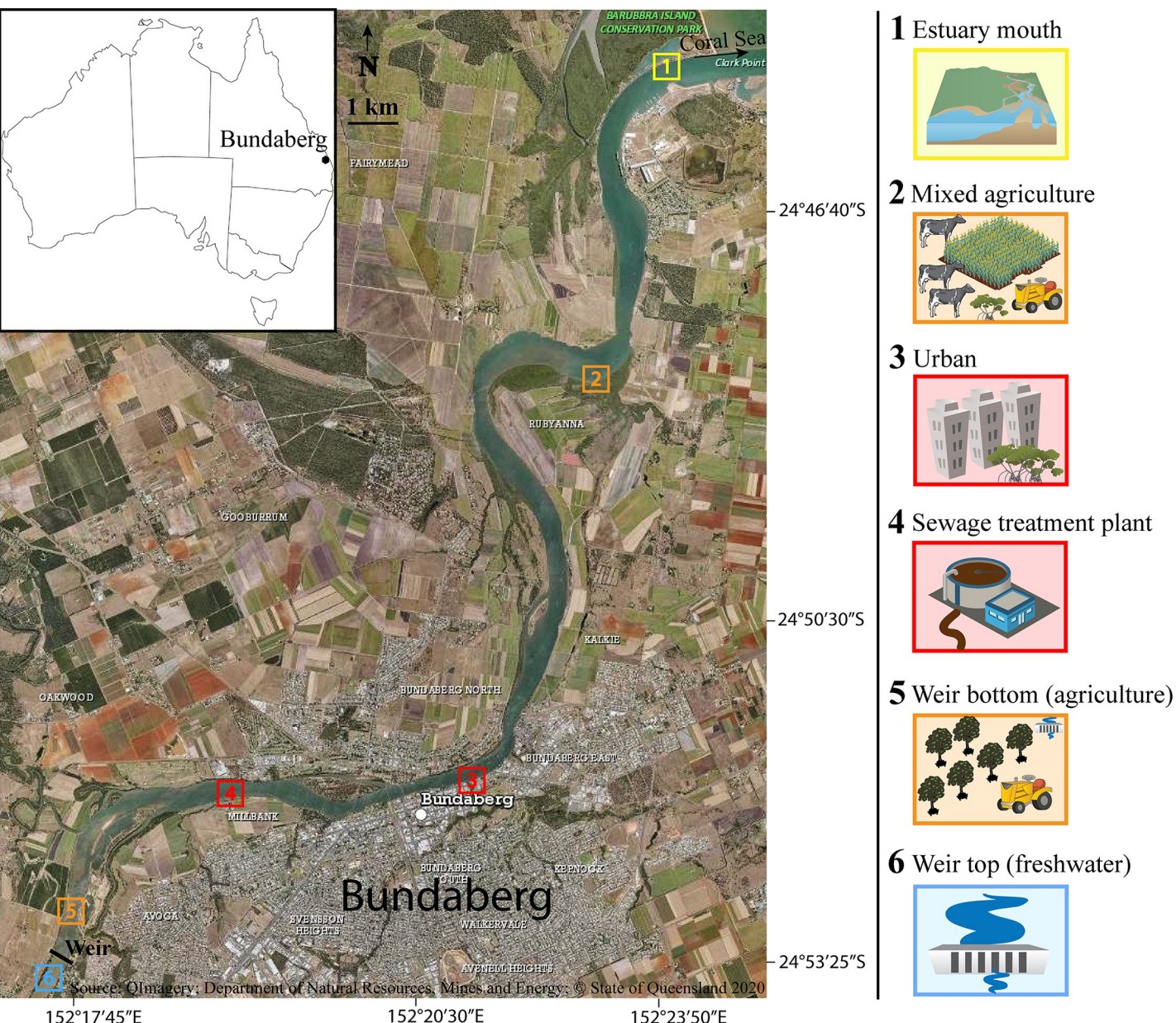

**Fig 1. Map of the subtropical Burnett River estuary, situated in the GBR catchment area (Queensland, Australia), depicting study sites along the land use gradient.** 1) Mouth of the estuary, mostly pristine environment; 2) Mixed agriculture site, sugar cane farming and livestock grazing; 3) Urban site within the city of Bundaberg, housing and industry; 4) Sewage treatment plant (STP), urban waste; 5) Bottom of the weir, macadamia farming; 6) Top of the weir, freshwater site.

drop during dry conditions due to the separation of estuarine brackish waters and upstream fresh water inputs. Small scale horticulture (predominantly macadamia farms) is situated to the west of the weir. Above and beyond the weir are low intensity agriculture and residential developments.

## 2.2 Sample collection

Sampling was undertaken between the 7th and 12th of January 2019 coinciding with the end of the dry season in the region. Most of the spatial sampling was conducted from a small boat

over three consecutive days on an ebbing tide, starting at the mouth at high tide. Discrete samples for microbiological analysis and nutrient analysis were taken at each site in conjunction with continuous water column measurement of GHG's and physicochemical parameters. Surface water GHG concentrations were sampled *in situ* using a closed-loop system, where a surface water fed air-water equilibrator was connected to a Cavity Ring-Down Spectrometer (CRDS, Picarro Gas Scouter) similar to that described by Maher et al. [48]. Due to restricted boat accessibility at Sites 5 and 6, discrete surface water samples were collected from the upstream sites and stored in gas-tight 6 L bottles leaving a headspace. Physicochemical parameters were also measured continuously *in situ* using a submersible multi-parameter sonde (YSI Exo2) measuring DO (% Sat), pH, salinity and temperature (˚C).

At each site, discrete surface water samples (~5 cm depth) were taken from the estuary, as well as shallow groundwater / pore water samples (80–100 cm depth) from the immediate adjacent shoreline (within 5 meters). Shallow groundwater wells were excavated in the intertidal zone, with each well purged dry three times to ensure fresh pore water was collected. Wells were left to recharge in between sampling rounds. With this method, the signal was integrated over the whole depth range. There was little change in water table depth along the estuary and sampling sites. For both surface water and pore water samples, DOC samples were filtered with 0.7 μm glass microfiber filters and stored in 40 ml borosilicate vials amended with saturated $H_3PO_4$ to stop further microbial processing. Ammonium, nitrate and sulfate samples were filtered using 0.45 μm cellulose acetate filters and stored in 10 ml polycarbonate vials (Thomas Scientific).

At all sites, individual microbial samples of both surface waters and pore waters were extracted by hand using fresh sterile examination gloves, filters and syringes to avoid cross-contamination. Microbial samples were filtered with 0.22 μm membrane filters (Isopore) and stored in sterile 1.5 ml polycarbonate vials (Eppendorf) pre-filled with DNAgard (Sigma) in a flow hood maintaining sterile conditions. As water was filtered until the membrane filters' capacity was saturated to maximize the yield of microbial tissue, total sample volumes varied from 41 ml to 129 ml for pore water samples and from 152 ml to 420 ml for surface water samples. In order to gather duplicates, sampling took place within 2 minutes of each other at each site in the same area (~1 m diameter), with the exception of Site 1 surface water samples, where the sampling vessel drifted due to tidal current and the duplicate had to be discarded. Hierarchical clustering was carried out with GENE-E (Broad Institute, MIT) using negative Pearson correlation metrics which indicated low dissimilarity between microbiological sample duplicates (S1 Fig).

## 2.3 Environmental sample analysis

DOC concentrations were determined using a total organic carbon analyser coupled with an isotope ratio mass spectrometer (Thermo Fisher Delta-V Plus) and a continuous flow system (Thermo Fisher ConFLo) with a precision of 0.02 mM [49]. Ammonium and nitrate concentrations were measured via flow injection analysis (Lachat 8500), with an analytical error of 5.1% for $NH_4^+$ and 6.2% for $NO_3^-$ [50]. Sulfate concentrations were determined via ion chromatography (Metrosep A Supp4–250 column and Metrosep RP2 guard column; eluent contained 2 mM $NaHCO_3$, 2.4 mM $Na_2CO_3$ and 5% acetone) with an analytical error of ~2% [51]. For discrete GHG samples, the gas-tight 6 L bottles were connected into a closed loop with the CRDS, with an inlet tubing bubbler used to encourage headspace equilibration. Each water sample was run for $\geq 2$ h or until gas concentrations between the water sample and headspace equilibrated. GHG and nutrient concentrations were averaged for each sampling location and error propagation for measurement uncertainties applied.

## 2.4 Microbiological sample analysis

DNA from microbial samples was isolated using the DNeasy PowerLyzer PowerSoil Kit (Qiagen). PCR amplification and sequencing were performed at the Australian Genome Research Facility using the primers and conditions outlined in S1 Table. Thermocycling was completed with an Applied Biosystem 384 Veriti and using Platinum SuperFi mastermix (Life Technologies, Australia) for the primary PCR. The first stage PCR was cleaned using magnetic beads, and samples were visualised on 2% Sybr Egel (Thermo-Fisher). A secondary PCR to index the amplicons was performed with TaKaRa PrimeStar Max DNA Polymerase (Clontech). The equimolar pool was cleaned a final time using magnetic beads to concentrate the pool and then measured using a High-Sensitivity D1000 Tape on an Agilent 2200 TapeStation. The pool was diluted to 5nM and molarity was confirmed again using a Qubit High Sensitivity dsDNA assay (ThermoFisher). Amplicons were quantified with a dsDNA fluorometry assay (Promega Quantifluor) after two rounds of PCR. Sequencing took place on an Illumina MiSeq (San Diego, CA, USA) with a V3, 600 cycle kit (2 x 300 base pairs paired-end) and a 25% PhiX spike-in to improve nucleotide diversity. A variation of the Illumina 16S metagenomics sequencing protocol was utilized for this purpose.

## 2.5 Data analysis

Paired-end reads were assembled by aligning the forward and reverse reads using PEAR (version 0.9.5) [52]. Primers were identified and trimmed before processing using Quantitative Insights into Microbial Ecology (QIIME) [53] USEARCH and UPARSE software [54, 55]. Sequences were quality filtered and full-length duplicate sequences were removed and sorted by abundance using USEARCH with singletons or unique reads in the data set discarded. Sequences were clustered before chimera filtering using the "rdp_gold" database as the reference. To obtain the number of reads in each operational taxonomic unit (OTU), reads were mapped back to OTUs with a minimum identity of 97%. Using QIIME, taxonomy was assigned with the Greengenes database [56].

  Community distributions for each site were visualized for the entire samples on phylum, class, order and family level using MEGAN software (MEtaGenome Analyzer) [57]. As sample volumes varied, the number of reads were normalized per 1 ml prior to analysis. Hill numbers were evaluated on family level to gauge the microbial diversity in all samples using Eqs 1–4.

$$H' = - \sum p_i \ln(p_i) \tag{Eq 1}$$

$$Hill_1 = \exp(H') \tag{Eq 2}$$

$$\gamma = \sum (p_i)^2 \tag{Eq 3}$$

$$Hill_2 = \frac{1}{\gamma} \tag{Eq 4}$$

  Hill$_1$ uses the exponential of the Shannon-Weaver index H' (Eqs 1 & 2). Hill$_2$ represents the reciprocal of the Simpson's index γ (Eqs 3 & 4). The number of reads found in the $i$th taxonomic family is depicted by p$_i$. Visualization of individual bar plots for physicochemical parameters, GHG and nutrient concentrations was carried out using Gnuplot software. Principal component analysis was carried out with Python software in the Spyder IDE with *pandas*, *scikit* and *matplotlib* libraries. To do this, data sets were standardized using StandardScaler() commands and projected into 2 dimensions using PCA() commands (i.e. pca.fit_transform()).

After transposition, resulting data sets were individually plotted into the principal subspace of the first 2 principal components (PCs) with *pyplot*.

## 3 Results

### 3.1 Physicochemical parameters, nutrients and greenhouse gases

Water temperature showed a maximum of 29.4˚C at the estuary mouth (Site 1) and a minimum of 27.4˚C at the mixed agriculture site (Site 2). This coincided with an average daytime air temperature of 28.6 ± 1.6˚C on sampling days and only one rainfall event (5.6 mm on the 9th of January, early in the morning). The estuarine salinity gradient decreased from 36.4 at the estuary mouth (Site 1), to 26.6 at the bottom of the weir (Site 5) and dropped to 0.2 at the freshwater site (Site 6) (Fig 2). Water column DO saturation ranged from 90.6% to 106.4% saturation at Sites 1, 4, 5 & 6 (Fig 2), with low DO (26.6–36.8%.) observed at the mixed agriculture and urban site (Sites 2 & 3). Pore water DO saturations are assumed to be close to 0% as seen in other study with largely impermeable sediment and high amounts of organic matter [58–60].

DOC concentrations ranged from 0.39 mM at the estuary mouth (Site 1) to 1.8 mM at the freshwater site (Site 6) and had a local maximum of 1.2 mM at the urban site (Site 3) (Fig 3). Ammonium concentrations continuously increased from 22.6 μM at the estuary mouth (Site 1) to 118.7 μM at the freshwater site (Site 6) with the exception of the bottom of the weir (Site 5) where low $NH_4^+$ concentrations were observed (4.1 μM). Nitrate concentrations were low at the mixed agriculture, urban sites and sewage treatment plant sites ($< 1.5$ μM; Sites 2, 3, 4) as well as at the freshwater side of the weir (Site 6) with only the bottom of the weir (Site 5) showing relatively high nitrate concentrations (46.5 μM). Sulfate concentrations in pore water were lowest at the freshwater site (0.03 mM; Site 6) and ranged from 1.7 to 3.3 mM at the other sites but did not follow the salinity gradient. Nutrient data from surface water samples revealed no relationship to relevant porewater communities (refer to S2 Fig).

$CH_4$ concentrations in the estuary varied widely along the salinity and land use gradient with concentrations ranging from 0.2 ± 0.01 μM at the estuary mouth (Site 1) to 2.68 ± 0.2 μM at the freshwater site (Site 6) (Fig 4). Downstream of the weir, maximum $CH_4$ concentrations were observed in the water column at the urban site (Site 3; 0.51 ± 0.04 μM) and the sewage treatment plant (Site 4; 0.66 ± 0.11 μM). $CO_2$ concentrations in the water column were highest at the urban site (Site 3; 212 ± 14 μM) and ranged from 14 μM to 54 μM at all other sites (Fig 4). Both $CH_4$ and $CO_2$ concentrations were only measured in the water column at a fixed depth (0.5–0.7 m) due to the sampling design. Additional oxidation of gas fluxes prior to reaching the water column sampling location thus need to be considered for pore water processes.

### 3.2 Prokaryotic community composition and diversity

There were notable differences in community distribution on phylum level between the surface and pore water samples within each site, as well as trends along the estuarine land use gradient (Fig 5). *Cyanobacterial* OTUs mainly occurred in surface water samples and were most prevalent at the mouth of the estuary with decreasing relative abundance along the salinity gradient up to the weir and were low at the mixed agriculture site and urban site (Sites 2 & 3) where DO was low. The functionally diverse *Proteobacteria* were abundant in all samples. Differences in community distributions were especially pronounced in the pore water samples with an increased relative abundance of *Firmicutes*, *Euryarchaeota* and *Chloroflexi* in the urban sites (Sites 3 & 4). The *Euryarchaeota* phylum includes a variety of methanogens with six of the seven known orders of methanogenic archaea detected in this study

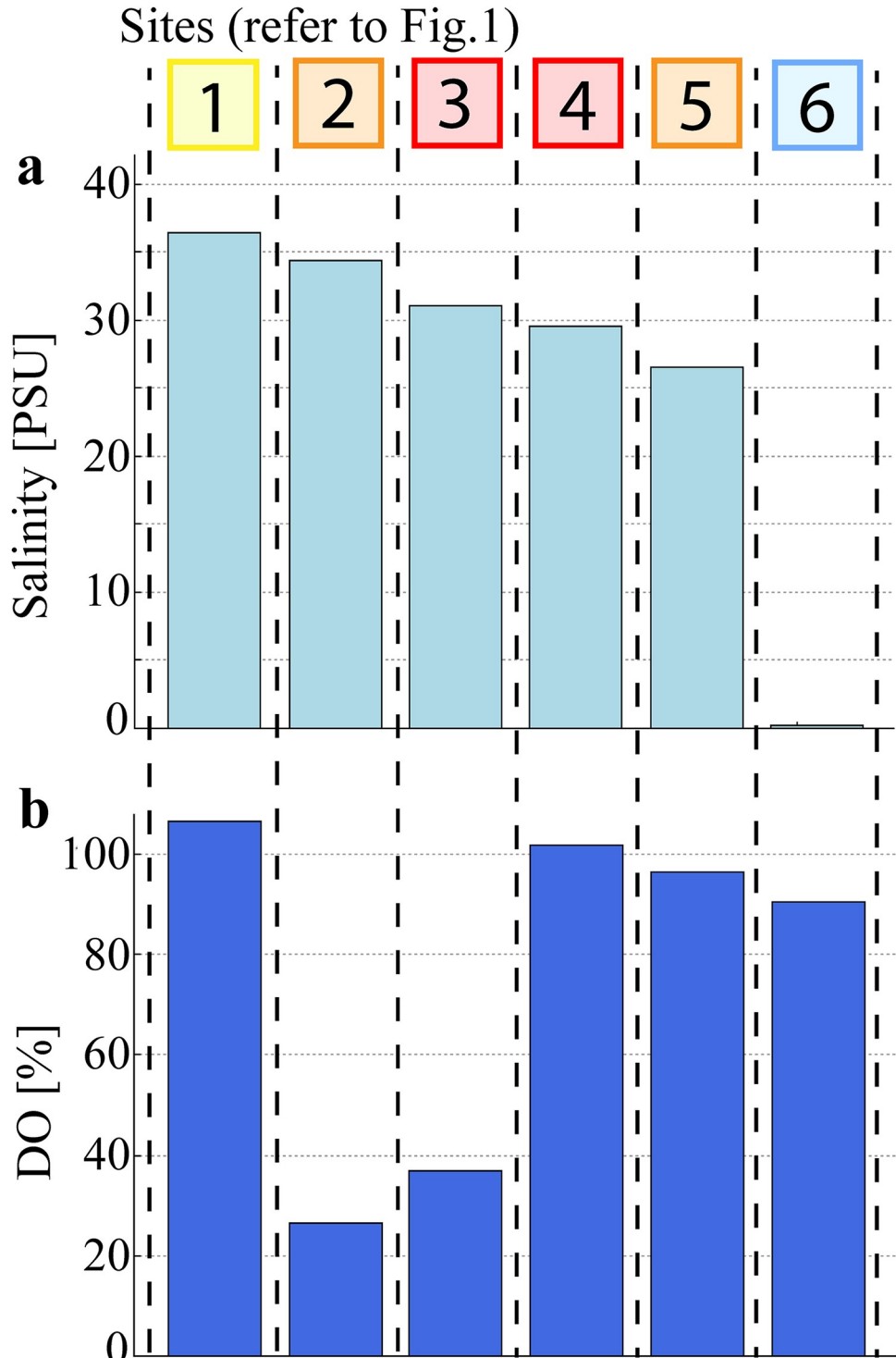

**Fig 2. Average physicochemical parameters in the water column of the Burnett River estuary for each site.**

(*Methanobacteriales*, *Methanocellales*, *Methanococcales*, *Methanomassiliicoccales*, *Methanomicrobiales* and *Methanosarcinales*) [19, 61] (refer to Fig 6). The Chloroflexi phylum was dominated by the dehalogenating order *Dehalococcoidales* which have been previously depicted as

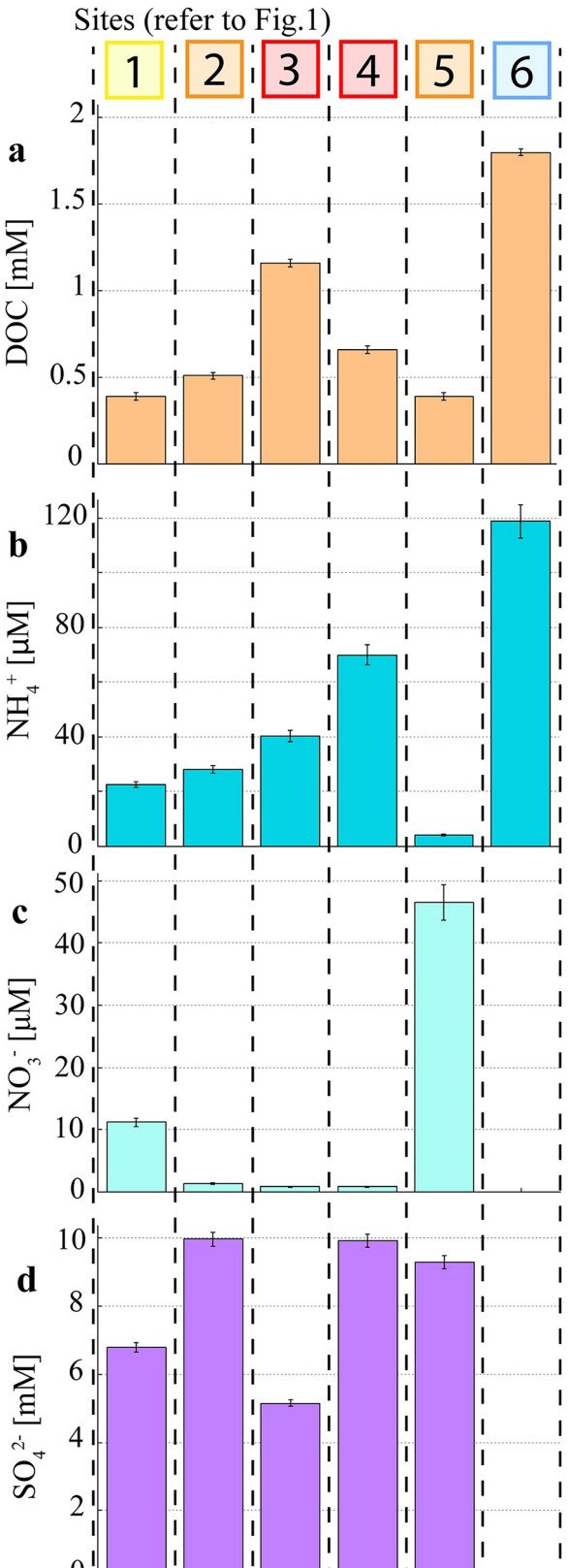

**Fig 3. Pore water nutrient concentrations at the different sites along the Burnett River estuary.**

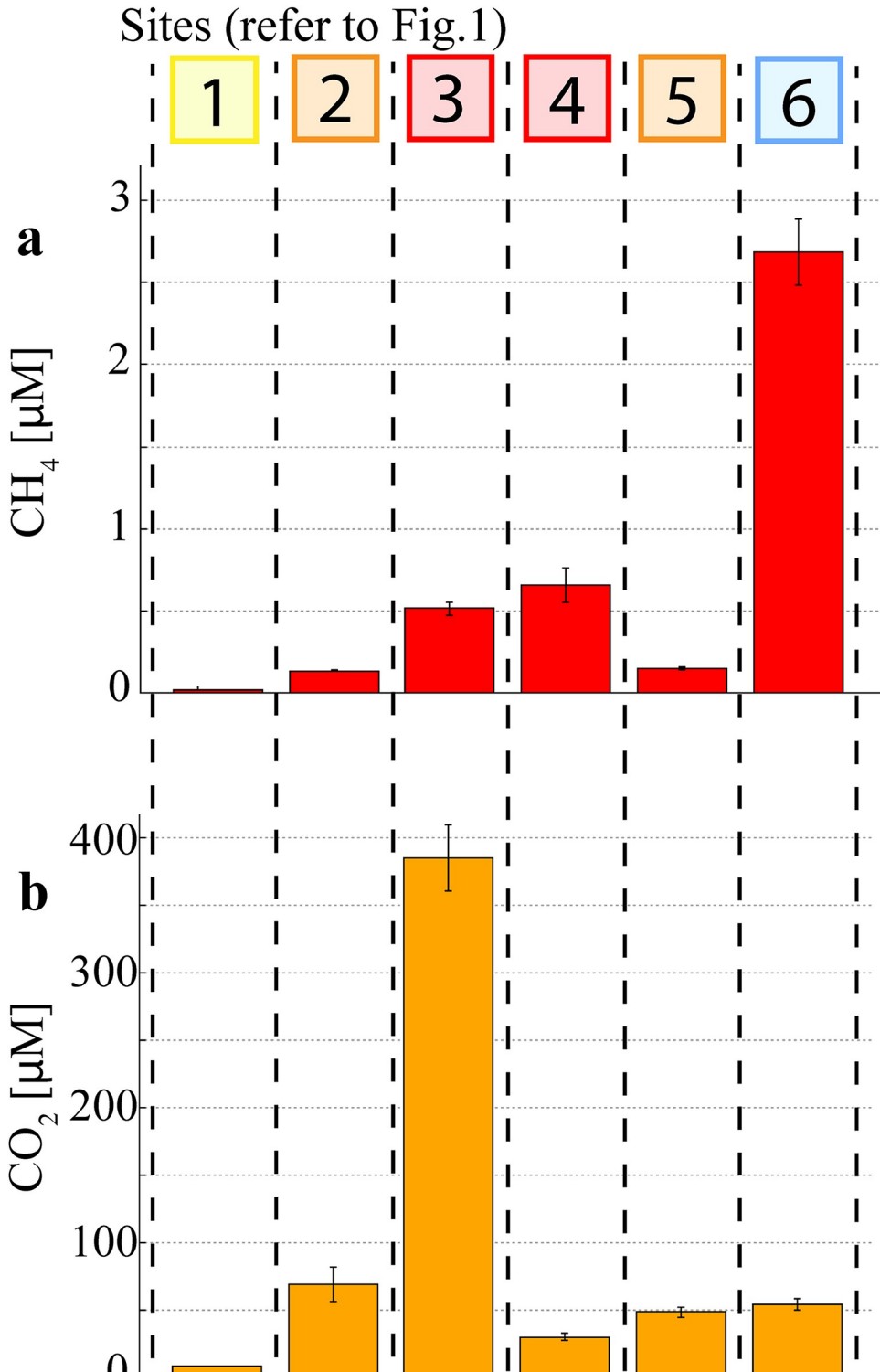

**Fig 4. Water column greenhouse gas concentrations along the Burnett estuary land use gradient.** Note: Error bars represent ± SD of averaged data from CRDS.

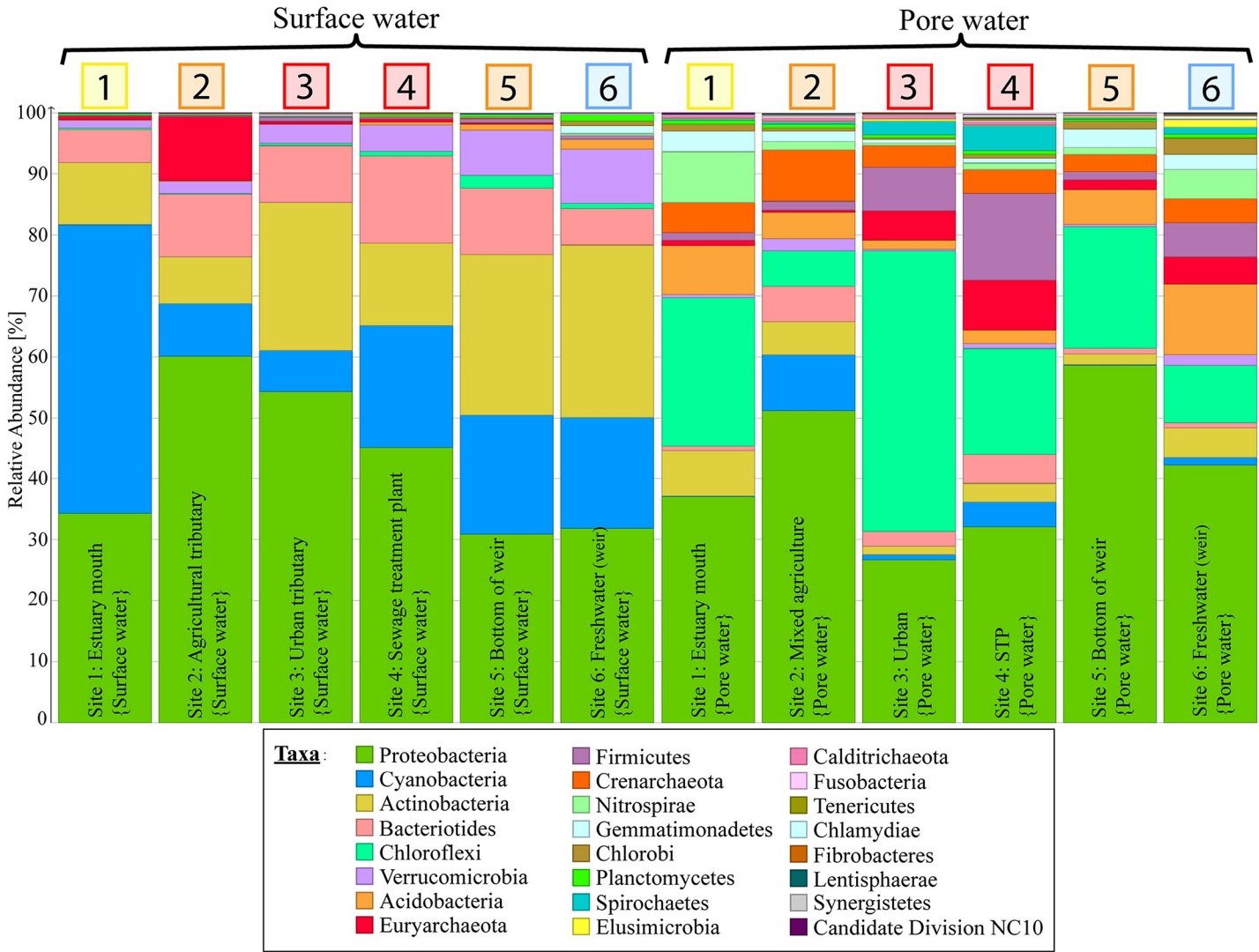

**Fig 5. Community bar plot of prevalent prokaryotic phyla in the Burnett River estuary.** Relative abundances are depicted for each phylum per sampling site. Phyla with an abundance of <0.1% in all individual samples were discarded.

bioremediation bacteria in polluted environments [62–64]. Notably, the total number of dsDNA detected per ml varied between samples and ranged from 5,027 at the urban site (Site 3) to 12,346 at the freshwater site (Site 6) in surface water samples and between 6,967 at the bottom of the weir (Site 5) and 14,743 at the mixed agriculture site (Site 2) in pore water samples (S2 Table).

In surface water samples, the diversity indexed by $Hill_1$ (more weight on OTU richness) and $Hill_2$ (more weight on OTU evenness) was highest at the urban site (Site 3; $Hill_1$ = 75.2, $Hill_2$ = 12.2) and lowest at the estuary mouth (Site 1) ($Hill_1$ = 4.9, $Hill_2$ = 1.7) (Table 1). Average diversity in pore water samples was overall higher ($Hill_{1,AVG}$ = 72.6 ± 42.2, $Hill_{2,AVG}$ = 10.2 ± 5.6) than surface water samples ($Hill_{1,AVG}$ = 39.9 ± 22.3, $Hill_{2,AVG}$ = 6.7 ± 2.5) with the highest values at the sewage treatment plant (Site 4; $Hill_1$ = 123.4, $Hill_2$ = 17.7) and the lowest diversity observed at the bottom of the weir (Site 5; $Hill_1$ = 22.3, $Hill_2$ = 3.6) (Table 1). Diversity trends followed the land use gradient with more diverse communities occurring in the

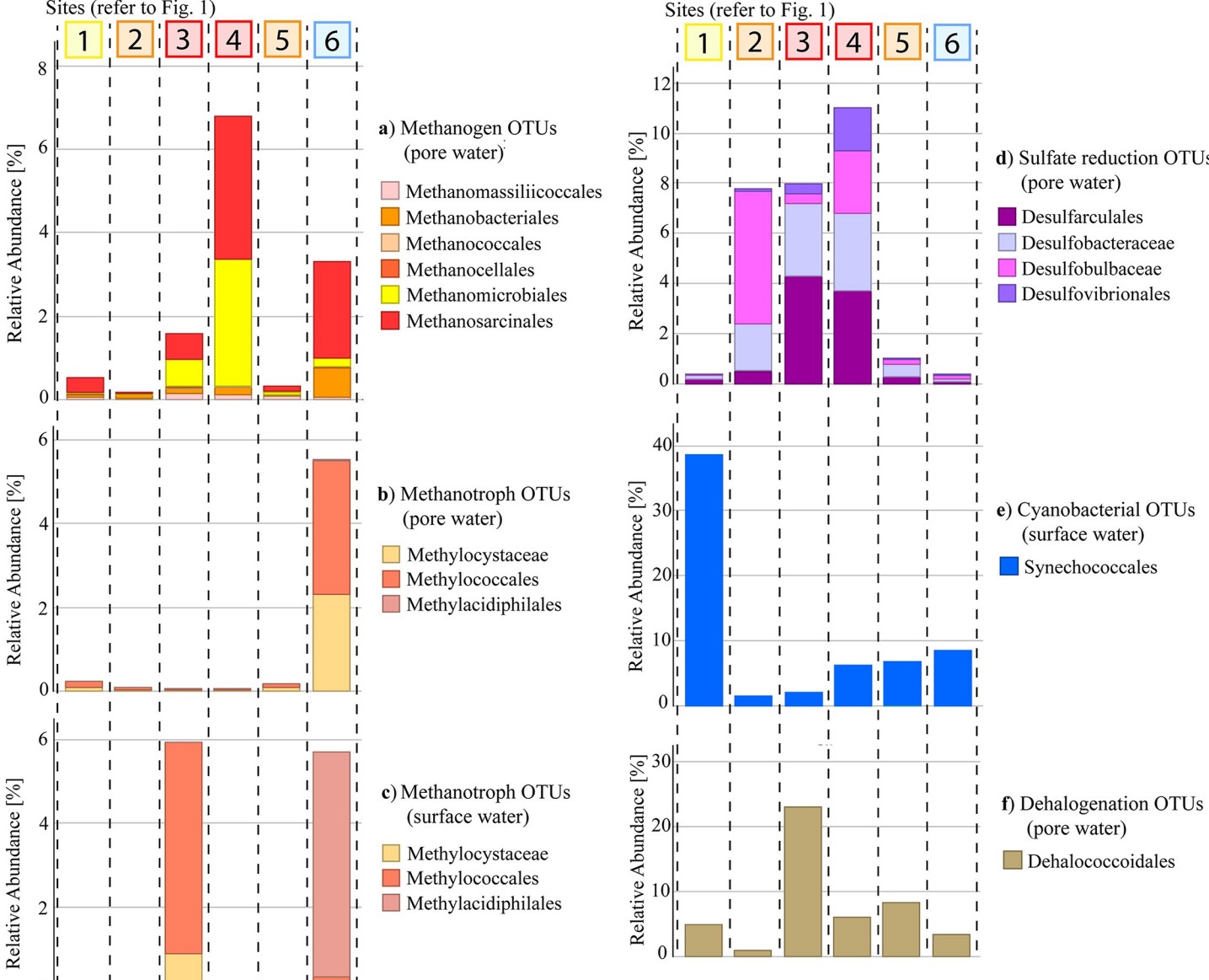

**Fig 6. The relative abundances of select environmental OTUs along a land use gradient in the Burnett River estuary.** Functional groups were combined in the stacked bar plots.

agricultural and urban sites (Sites 2 & 3) around the city centre as well as at the freshwater side of the weir (Site 6). Exceptions to this trend were found in the diversity of prokaryotic communities in STP (Site 4) surface water samples and urban (Site 3) pore water samples.

**Table 1. Hill numbers of diversity indices calculated for all sites in surface and pore water.** $Hill_0$ represents the detected number of OTUs for each sample.

| Site | Surface water samples | | | | | | Pore water samples | | | | | |
|---|---|---|---|---|---|---|---|---|---|---|---|---|
| | 1 | 2 | 3 | 4 | 5 | 6 | 1 | 2 | 3 | 4 | 5 | 6 |
| $Hill_0$ | 92 | 111 | 151 | 107 | 128 | 129 | 125 | 135 | 119 | 135 | 121 | 136 |
| $Hill_1$ | 4.9 | 19.1 | 75.2 | 28.7 | 41.7 | 69.6 | 42.9 | 117.92 | 26.1 | 123.4 | 22.3 | 103.0 |
| $Hill_2$ | 1.7 | 4.3 | 12.2 | 6.7 | 8.3 | 7.0 | 6.26 | 15.3 | 3.7 | 17.7 | 3.6 | 14.2 |

### 3.3 Environmental OTUs

Prokaryotic pore water communities shifted towards methanogenic archaea along the land use gradient (Fig 6). Methanogenic archaea in anoxic pore water samples showed a higher relative abundance at the urban site (Site 3) and sewage treatment plant site (Site 4) as well as the freshwater site (Site 6). Relative abundances of pore water methanogenic archaea ranged from 0.2% at the mixed agriculture site (Site 2) to 7.8% near the sewage treatment plant (Site 4). The relative abundance of methanogenic archaea in pore water samples from the freshwater site (Site 6), which corresponded to the highest $CH_4$ and DOC concentrations (Figs 3A & 4A), was only 3.3%, but the total number of reads from freshwater site (Site 6) pore water samples was 38% higher than at the urban site (Site 3) (see S2 Table). Methanogenic archaea detected in surface water were below 0.01% (or 25 reads) and thus not further considered. The proportion of methanotroph OTUs was highest at the freshwater site (Site 6) in surface and pore water samples (relative abundances of 5.7% and 5.6% respectively) and in urban (Site 3) surface water samples (relative abundance of 5.9%; lowest total number of reads). Freshwater (Site 6) surface water samples were the only ones dominated by the thermoacidophilic methanotroph order *Methylacidiphilales*, with only a low fraction of type I or type II methanotrophs. The relative abundance of methanotrophs was below 0.5% at all other sites. All detected methanotrophic prokaryotes belonged to the bacteria kingdom with no ANME (anaerobic methanotrophic archaea) found in any of the samples. Denitrifying anaerobic methane oxidation (DAMO) bacteria belonging to the *Candidate Division NC10* phylum were detected in Site 1 pore water samples but had a low relative abundance of ~0.3%.

Coinciding with relatively high concentrations of DOC ($> 0.5$ mM) in pore water at Sites 2, 3 and 4, the relative abundance of sulfate reducing OTUs ranged from 7.8% to 10.5% and was below 1% at all other sites. The relative abundance of the dehalogenation OTU *Dehalococcoidales* (accounting for 99.9% of *Chloroflexi*) was highest at the urban dominated Site 3 (22.9%) and ranged from 1.0% to 8.1% at all other sites. *Synechococcales* was the most abundantly occurring order of the *Cyanobacteria* phylum (accounting for 99.8% of *Cyanobacteria*) in surface water samples and had the highest relative abundance (39.1%) at the mouth of the estuary (Site 1) and the lowest at the mixed agriculture and urban sites (Sites 2 & 3; 1.5% and 2% relative abundance respectively) where DO saturation was the lowest.

## 4 Discussion

Prokaryotic communities showed distinct changes in taxonomic profiles between land use types as well as along the estuarine salinity gradient within the Burnett River estuary. In particular, we found the highest abundance of methanogenic archaea in the urban and STP sites (Sites 3 and 4) of the estuary (Fig 6), which corresponded to elevated $CH_4$ and $CO_2$ concentrations (Fig 4). In other urbanised catchments, the presence of elevated concentrations of DOC coupled with an absence of thermodynamically favourable electron acceptors has been shown to facilitate the proliferation of methanogenic archaea [14, 65, 66]. This implies that increased inputs of organic matter due to catchment urbanisation and modification may lead to an increase in methanogenesis and GHG production [11, 12].

Principal component analysis (Fig 7) supports the strong relationship between pore water methanogens and DOC concentrations, as well as water column $CO_2$ concentrations, which can be used as energy sources for $CH_4$ generation [14, 28]. The abundance of methanogens however did not gradually increase moving upstream along the salinity gradient as observed in previous studies [41, 67, 68]. Conversely, urban and sewage related high DOC concentrations and scarcity of thermodynamically favourable electron acceptors (e.g. DO, $NO_3^-$) in the pore water appear to cause a stark shift in prokaryotic community compositions, with the highest

abundance of methanogens found adjacent to the STP (Site 4; refer to Fig 6). Highly labile DOC compounds from sewage related run-off are subject to rapid microbial degradation and have been previously shown to increase methane abundance [69]. This suggests that land use differences along the estuary may have a stronger effect on prokaryotic community structure than the salinity gradient alone. Conversely, aerobic conditions in the surface water (DO concentrations of 26.6% to 106.4%; Fig 2) limit methanogen communities to the anoxic pore water.

In the archaeal communities, the relative abundances of detected methanogenic orders belonging to the *Euryarchaeota* phylum varied between sites. For example, there was a 5 to 15 times lower relative abundance of *Methanomicrobiales* and a 4 to 8 times higher relative abundance of *Methanobacteriales* in the freshwater site (Site 6) compared to the urban and STP sites (Sites 3 & 4). The metabolically diverse *Methanosarcinales* had a high relative abundance in pore water samples at all sites where methane concentrations were high (between 41% and 68% of total methanogen communities at Sites 3, 4 and 6). Differences in the relative abundances of these *Euryarchaeota* may have contributed to changes in $CH_4$ concentrations at the sites with a high relative abundance of methanogens. The PCA plot reveals that *Methanobacteriales* and *Methanocellales* follow the DOC gradient, whereas *Methanococcales*, *Methanomassiliicoccales*, *Methanomicrobiales* and *Methanosarcinales* more closely follow the $CO_2$ gradient (Fig 7). The methanogen OTUs that follow the $CO_2$ gradient show clustering with sulfate reducing OTUs as well as the with the urban and STP sites (Site 3 & 4) and could thus be the main drivers of the observed local maxima in GHG production at these sites.

There was a considerably higher abundance of the order *Methanobacteriales* in the freshwater (Site 6) samples, where the highest $CH_4$ concentrations were seen (2.68 μM). Conversely, the *Methanomicrobiales* order had the highest abundance at the sewage treatment plant (Site 4), where the highest $CH_4$ concentrations along the estuarine salinity gradient were observed (0.66 μM; Sites 1–5) (Fig 4). As there is almost no sulfate (Fig 3) and a low relative abundance of sulfate reducers (Fig 6) present at the freshwater site (Site 6), a lack of competition for hydrogen produced by fermenters in the anoxic pore water could explain higher methane production rates of methanogens which commonly use $H_2$ as an electron donor [34, 70]. Further, differences in methane production rates between orders could contribute to the observed differences between $CH_4$ concentrations at the freshwater site (Site 6) and the other sites (Sites 1–5). The quantification of methane production rates of different methanogenesis taxa is still lacking in the literature owing to difficulties in culturing most of these microorganisms [6, 19, 71]. To narrow down contributions of individual methanogenic OTUs, amplicon sequencing data and in situ $CH_4$ concentrations could be combined with further microbiological methodology like shotgun metagenomics or transcriptomics to gauge the abundance and/or expression of methanogenesis genes (e.g. the *mcrA* gene) in the environment.

Within the more impacted urban and STP sites (Sites 3 and 4), there was a co-occurrence of high relative abundances of sulfate reducing bacteria and methanogenic archea in pore water, pointing to a potential syntrophic relationship between the two groups. Sulfate reducing bacteria have a high affinity to hydrogen and acetate and readily outcompete methanogens for these common electron donors [40, 70]. A high loading of labile DOC compounds in the substrate however can limit competition and allow for co-existence of sulfate reducing bacteria and methanogenic archaea [34, 40, 72]. The high concentration of DOC in the urban environment around Bundaberg (i.e. urban/STP, Sites 3 & 4) thus seems to prevent sulfate reducers and methanogens from outcompeting each other and enable substantial communities of both functional groups to co-exist and form syntrophic relationships. These results also point to a low concentration of humic substances in the DOC, which have been shown to inhibit both

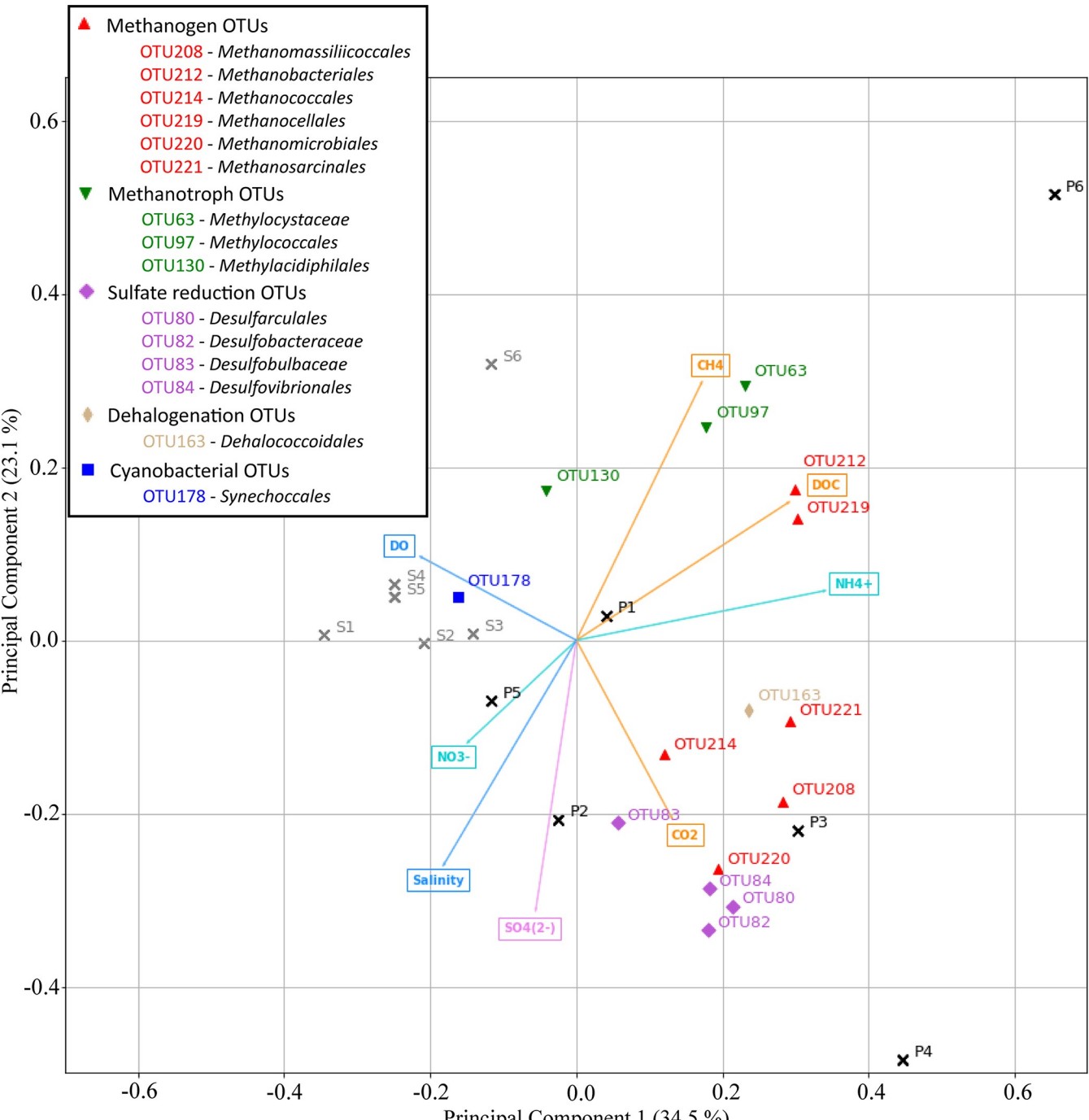

**Fig 7. PCA biplot depicting variations in environmental parameters (nutrient, greenhouse gas and physicochemical).** OTUs included in the scatter plot are specified on the upper left. Surface water samples for Sites 1–6 are shown as grey crosses with the suffix 'S' and pore water samples are depicted as black crosses with the suffix 'P'.

methanogenesis and sulfate reduction as well as serve as an electron acceptor for ANME (absent in our samples; see section 3.3) [73–75].

The study also found high $CO_2$ concentrations in the most impacted sites (i.e. urban/STP, Sites 3 & 4) which can fuel microbially mediated methane generation. Sulfate reducers can

account for almost 100% of total $CO_2$ (via production of bicarbonate) in polluted mangrove sediments [5, 76]. This $CO_2$ can fuel methanogenesis if methanogens find a favourable substrate to proliferate [14, 32, 77]. That is reflected by a high abundance of both sulfate reducing bacteria (7.8% to 10.5%) and methanogenic archaea (7.8% at Site 4) as well as high concentrations of $CH_4$ (0.51 to 0.66 μM) and $CO_2$ (212 μM at Site 3) in the urban and STP sites (Sites 3 & 4) of the estuary. Urbanized estuaries could thus facilitate a GHG-fuelled microbial loop between methanogenic archaea and sulfate reducing bacteria that drastically increases $CH_4$ and $CO_2$ emissions in these environments. However, the exact mechanism of interplay between individual OTUs which is resulting in increased GHG concentrations along estuarine land use gradients was not determined.

A lack of $NO_3^-$ was detected in the more impacted sites where high relative abundances of methanogens and sulfate reducers were observed in pore water (urban/STP, Sites 3 & 4). Conversely, $NO_3^-$ concentrations were high (46.5 μM) in the pore water at the bottom of the weir (Site 5; inland macadamia agriculture) where low concentrations of $CH_4$ (< 0.15 μM) and $CO_2$ (< 0.03 μM) were detected. PCA analysis shows the inverse relationship between $NO_3^-$ and $CH_4$ dynamics, highlighting the inhibition of methanogenesis at the bottom of the weir (Site 5) (Fig 7). Relatively low $NH_4^+$ concentrations (4.1 μM) were also observed in the high $NO_3^-$ containing pore water samples downstream of the weir (Site 5). Anaerobic ammonium oxidation (anammox) can build up $NO_2^-$, an intermediate in the production of $NO_3^-$ that is toxic to methanogens [38, 78]. In previous studies, high $NO_3^-$ concentrations have also been linked to the inhibition of $CH_4$ generation due to denitrification bacteria outcompeting methanogens [35, 38, 79]. Fang et al. [79] showed in an upflow anaerobic sludge blanket reactor which was treating wastewater containing phenol, that methanogenesis only occurred after all denitrification had been carried out in the substrate and only if organic material (measured via chemical oxygen demand to $NO_3^-$ –N ratios) was still available after fuelling the denitrification bacteria.

We found a concomitant relationship between the high abundance of methanotrophs and $CH_4$ at the urban and freshwater sites (Site 3 and 6). Pore water methanotrophs have distinct differences in phylogeny, with the recently discovered *Methylacidiphilales* order representing a majority of taxa at the freshwater site (Site 6) while being largely absent at all other sites. The potential existence of distinct microenvironments entailing different pH could explain the prevalence of acidophilic *Methylacidiphilales* at the freshwater site [80, 81]. Varying rates of $CH_4$ oxidation in different taxa could be hypothesized to contribute to observed differences in $CH_4$ concentrations. However, methanotrophic microorganisms have not yet been well characterized [82]. Additional research focussing on environmental methanotrophs could include the isolation of environmental samples containing a high proportion of a single methanotroph OTU (e.g. *Methylacidiphilales*). The abundance of these methanotrophs could subsequently be determined together with $CH_4$ oxidation rates in the samples which could allow for a quantitative link between rates and OTUs. It is also noted, that total abundance measurements via PCR-based methods are generally error-prone and further research could be improved by employing additional quantification assays like flow cytometry [83–86].

At the mouth of the estuary, the relative abundance of cyanobacterial OTU (*Synechococcales*) was 4 to 20 times higher than in the other surface water samples (38%; Site 1). This is likely due to oceanic *Synechococcus sp.* dominated communities transported into the estuary during each tide as has been reported previously [87]. Relative abundances of photosynthetic bacterioplankton in surface water communities of the estuary were closely associated with physicochemical gradients [21, 88]. The dehalogenation OTU *Dehalococcoidales* had a strong inverse relationship to *Cyanobacteria* and the physicochemical gradients in the PCA plot (Fig 7). This occurred prominently at the urban site (Site 3) and almost exclusively in pore water

samples, suggesting its potential use as a marker taxon for urban and industrial run off. *Dehalococcoidales* may also play an important role in the bioremediation of harmful compounds (e.g. Chlorobenzenes and chloroethylenes) discharging into the potentially sensitive GBR marine park [62, 64]. Torlapati et al. [64] revealed an acceleration of dechlorination and bioremediation when bacterial growth of *Dehalococcoides sp*. was increased in an artificial groundwater aquifer.

The main driver of GHG cycling prokaryote distributions was likely the carbon biogeochemistry along the land use gradient, and especially in the urban, STP and freshwater environments (Sites 3, 4 & 6). Most of the investigated OTUs and urban pore water samples are aligned with the carbon compounds (Fig 7). This is also apparent in individual bar plots of prokaryotes (Fig 6A, 6D and 6F) and carbon compounds (Figs 3A, 4A and 4B) which all have local maxima at the sites with a higher urbanisation (urban/STP, Sites 3 & 4). Diversity indices of the entire microbial communities also showed increased values at highly impacted sites rather than following physicochemical gradients (Table 1). This implies a lowered significance of the physicochemical parameters which are commonly depicted as the main driving forces of microbial community compositions in estuarine environments [21, 22].

Overall, this study highlights the relationship between the distribution of microbial communities and GHG dynamics along an estuarine land use gradient. Most microbes are restricted to specific niches in the environment they live in [32, 89, 90]. Physicochemical parameters like DO and salinity impact these niches as described in previous studies [21, 23] and correspondingly evoked changes in microbial communities along the Burnett River estuary. However, the shift in community compositions along the Burnett River estuary seemed to be highly influenced by urban land use zones which had a general scarcity of nutrients but elevated concentrations of carbon compounds. The changes in land use resulted in distinct ecological niches that likely facilitated the proliferation of methanogenic archaea and sulfate reducers as well as leading to considerable increases in GHG concentrations. More research on how land use mediates shifts in environmental parameters and microbial communities within these ecological niches needs to be carried out to better understand GHG emission and nutrient fluxes to sensitive ecosystems such as the GBR. Additionally, specific OTUs and genes involved in GHG dynamics in these vulnerable coastal ecosystems need to be characterized further.

## Supporting information

**S1 Table. Thermal cycling lengths with temperatures and primers used for the PCR procedure.**
(DOCX)

**S2 Table. Total number of dsDNA per ml of each sample, quantified by fluorometry (see section 2.3).**
(DOCX)

**S1 Fig.** Hierarchical clustering plot using negative Pearson correlation metrics for a) pore water samples (sampling sites numbered P1 –P6) and b) surface water samples (sampling sites numbered S1 –S6); Duplicates are denoted after sample numbers in blue. Dissimilarity heights ($\geq 0.01$) between duplicates are shown in green.
(TIF)

**S2 Fig. Surface water nutrient concentrations at the different sites along the Burnett River estuary.** No direct links to relevant prokaryotic pore water communities.
(TIF)

**S1 Data.**

(CSV)

**S2 Data.**

(CSV)

**S3 Data.**

(CSV)

## Author Contributions

**Conceptualization:** Sebastian Euler, Luke C. Jeffrey, Douglas R. Tait.

**Data curation:** Sebastian Euler, Damien T. Maher, Douglas R. Tait.

**Formal analysis:** Sebastian Euler.

**Funding acquisition:** Sebastian Euler, Douglas R. Tait.

**Investigation:** Sebastian Euler, Luke C. Jeffrey, Derek Mackenzie.

**Methodology:** Sebastian Euler, Luke C. Jeffrey, Damien T. Maher, Douglas R. Tait.

**Project administration:** Sebastian Euler, Douglas R. Tait.

**Resources:** Sebastian Euler, Luke C. Jeffrey, Damien T. Maher, Derek Mackenzie.

**Software:** Sebastian Euler, Luke C. Jeffrey.

**Supervision:** Damien T. Maher, Douglas R. Tait.

**Validation:** Sebastian Euler, Damien T. Maher, Douglas R. Tait.

**Visualization:** Sebastian Euler.

**Writing – original draft:** Sebastian Euler.

**Writing – review & editing:** Luke C. Jeffrey, Damien T. Maher, Douglas R. Tait.

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
