## [Decision Letter · Decision Letter 0]

16 Jun 2020

PONE-D-20-09256

Shifts in methanogenic archaea communities and methane dynamics along a subtropical estuarine land use gradient

PLOS ONE

Dear Dr. Euler,

Thank you for submitting your manuscript to PLOS ONE. After careful consideration, we feel that it has merit but does not fully meet PLOS ONE’s publication criteria as it currently stands. Therefore, we invite you to submit a revised version of the manuscript that addresses the points raised during the review process.

ACADEMIC EDITOR: 

Changes of methodological description on sampling pore water, presentation of pore water data, naming of sites, improvement of the discussion as pointed out by both reviewers are required for acceptance.Adding qPCR data is highly recommended. Please take special care in revising the text when talking about relative abundances according to the reviewers suggestions.  Please take special care about the description of the sampling procedure. The methods section could be improved in structure, making a distinction between sampling site, sampling procedure, environmental sample analysis, biological sample analysis and data analysis. Please add sample volumes. 

We look forward to receiving your revised manuscript.

Kind regards,

Ute Risse-Buhl, Ph.D.

Academic Editor

PLOS ONE

Journal Requirements:

2.  We note that[Figure 1 in your submission contain [map/satellite] images which may be copyrighted. All PLOS content is published under the Creative Commons Attribution License (CC BY 4.0), which means that the manuscript, images, and Supporting Information files will be freely available online, and any third party is permitted to access, download, copy, distribute, and use these materials in any way, even commercially, with proper attribution. For these reasons, we cannot publish previously copyrighted maps or satellite images created using proprietary data, such as Google software (Google Maps, Street View, and Earth). For more information, see our copyright guidelines: http://journals.plos.org/plosone/s/licenses-and-copyright.

Additional Editor Comments (if provided):

Both reviewers find strength in your study, but both highlight several aspects that must be clarified to make the study convincing. Please carefully consider all Reviewer comments and take special care about the missing qPCR data as highlighted by Reviewer 1, improving the description of methods and presentation of results, especially the data of pore water samples, and improve the discussion as highlighted by both reviewer.

Reviewers' comments:

Reviewer's Responses to Questions

**Comments to the Author**

1. Is the manuscript technically sound, and do the data support the conclusions?

Reviewer #1: Partly

Reviewer #2: Yes

2. Has the statistical analysis been performed appropriately and rigorously? 

Reviewer #1: Yes

Reviewer #2: Yes

3. Have the authors made all data underlying the findings in their manuscript fully available?

Reviewer #1: No

Reviewer #2: Yes

4. Is the manuscript presented in an intelligible fashion and written in standard English?

Reviewer #1: Yes

Reviewer #2: Yes

5. Review Comments to the Author

Reviewer #1: The presented manuscript, “Shifts in methanogenic archaea communities and methane dynamics along a subtropical estuarine land-use gradient”, addresses a valuable and understudied topic. As many environmental and anthropogenic factors may influence the microbial communities, with the focus on methane related microbes, these affect the abundance of these microbes and, as a result, the methane emissions.

The authors have carried out sampling and sequencing in order to find correlations for methanogen relative abundance and the environmental parameeters at different sites. It is a valuable dataset, and this manuscript has potential, although it needs some more experiments, clarity in the text, materials and methods and would certainly benefit from looking into the sequencing data in a bit more detail.

The study has one major drawback, which is the lack of qPCR data. Correlating the relative abundance of methanogens in samples with concentration measurements is not possible. When the absolute number of the total microbial community would be known, one can understand why the CH4 might be higher or lower at a site. If a percentage of the total is the only known number, one cannot compare sites in regards to abundance, whereas that would be recommendable.

Authors discuss methanogens and methanotrophs in terms of relative abundance, yet it remained unclear why specific parameters were present only for the surface water and not porewater, such as oxygen concentration? As the title tells, the focus is on methanogens, although the data is presented only for the surface samples and not for the porewater samples, whereas the sequencing data is available? Authors have looked into the relative abundance of methanotrophs, although there was no discussion on anaerobic archaeal methanotrophs. These belong to the same order as methanogens, so the relative abundance of methanogenic orders includes microbes carrying out the reverse reaction. Suggestion is to add this information to the manuscript as then it would be complete in the methane cycle, looking at both microbial groups, methane producers, and consumers over environmental gradients as well as polluted versus more pristine sites.

Overall comment for making it easier to read, authors use different terminology throughout the manuscript for the sites, either site 1-5, or urban and agricultural site or brackish and estuary mouth. Could the authors please choose one, as it would be easier to understand the differences between these sites?

The data presentation needs polishing, as the thermal cycling of the PCR could be in the supplements and the environmental parameters for all sites shown in the manuscript itself.

Manuscript will benefit if the figures for methanogens in the porewater samples are shown the same way as for the surface water.

The sequencing data should be deposited so that the analysis could be repeated on the raw data and it would be publicly avaliable.

Specific comments:

Abstract:

It reads as DOC concentrations were considerably higher at the freshwater site-1800uM than in the estuarine sites. Could the authors add these values?

Materials and methods:

Q1: Did the authors consider adding qPCR data, as the importance of relative abundance in samples depends on absolute numbers. In a sample with lower microbial abundance, the fraction assigned as methanogens does not count and remains challenging to correlate directly with the methane concentrations.

Q2: 170-172: could the authors clarify how the porewater samples were taken? Was the surface water taken from the top~5cm, and immediately porewater at the same location from the bottom? Or sampled at a different location, closer to the shore?

How much did the water column hight vary along the estuary and the sampling sites?

Q3: L181: Were the tubes pre-filled with DNAgard?

Q4: L182: What is the relevance of the 2 min interval in sampling? For replication, to ensure the sampling of the community and water samples at the exact time point?

Q5: L200: Could the authors clarify these steps or refer to a protocol supplied by the sequencing facility.

L214: The sentence does not read well, replace clustered followed with clustered before/prior chimera filtering.

L215: Check for double spacing.

Q6: L219-220: What do authors mean by sampling volume?

Results:

Q1: The presented data for oxygen, methane etc., is presented for surface water. Is there the same data available for porewater samples? It would be easier for the reader if both datasets are presented within the manuscript and not as a figure in the manuscript and a table with additions in the supplementary table.

Q2: The presented prokaryotic community data is discussed in section 3.3. Is that for porewater or surface water?

Q3: On figures is presented the methanogen relative abundance in the porewater, could the authors add the same figure/data for surface water?

Q4: Methanogen relative abundance is high at the sites with higher methane concentrations as well as higher DOC concentration. However, methanogens are considered mostly as anaerobic microorganisms. Could the authors comment on these correlations, especially with oxygen?

L239-240: It is subjective. For some microbes, a 2C difference might not be small.

L242: Remove minor.

L251: In the abstract, the values are presented on the micromolar scale. For consistency throughout the manuscript, please use the same units. It makes it easier for the reader.

Q5: L321-323: Could the authors add the same data for methanogens?

Discussion:

Q1: In case there was a high relative abundance of methanogens at sites 3 and 4, as well as more DOC, what feeds this reaction? The discussion could go more in depth in this topic, as humic substances are part of DOC, and have shown to suppress/enhance methanogenesis, whereas simultaneously serve as the electron acceptor for anaerobic methanotrophs (ANME archaea).

Q2: As there is a difference between sites 3 and 4 in the oxygen content, although not much lower methane concentrations. One could think that anaerobic/less oxic conditions would make methanogens thrive. How can this be explained or how less oxic conditions affect the distribution of the overall community? Would more taxonomic groups of fermenting microbes increase?

Q3: Lines 375-377, it is not possible to relate a percentage of sequencing data with the methane concentration. One would need to know the absolute number or retrieve that from the absolute copy numbers quantified with the same primers with what the sequencing was carried out. Could the authors explain how concentration can be correlated with a percentage of microbes in this dataset?

Q4: The authors found Methylacidiphilales order, yet these microbes are mainly found in very acidic sites with pH down to 2. In extreme conditions with high temperatures, what could be the role of these in an estuary system?

Q5: Has this order of Methylacidiphilales been found in previous studies in estuaries? The found sequences in this order clustered all together or showed variations at lower levels? Could the authors go down to the genus level?

Q5: Could the authors include a discussion on anaerobic methanotrophs in case these were present in the data? It would be beneficial to find the literature on ANME archaea, look into salt and freshwater differences, and their contribution.

Reviewer #2: The manuscript by Euler et al entitled “Shifts in methanogenic archaea communities and methane dynamics along a subtropical estuarine land use gradient” describes the effect of changes in physiochemical, nutrient, and greenhouse gas distributions on the microbial communities in an estuary. Specifically, anthropogenic altered environments were compared to more pristine sites within the estuary. DOC, methane, O2, SO4, etc. were measured in surface in pore water samples and correlated with microbial identification and abundances using 16s rRNA. Overall, the manuscript is clear and concise. The conclusions are largely supported by the data. There are no apparent technical issues. However, the results are largely confirmatory of previous studies and expected; environments with higher DOC and scarce thermodynamically favorable electron acceptors have increased methanogenesis (more methanogens), which in turn can support methanotrophs. Urban and sewage treatment plant sites would be expected to result in increased DOC etc. This limits the significance of the study.

Specific Points:

1. Abstract lines 37-39: This sentence is confusing as written. It implies a comparison SRBs to electron acceptors. Possible changing “whilst” to “consistent”

2. Lines 98-105 (and other sections): As written, this paragraph implies methane consumption is only performed by bacteria, which of course is not true. Anaerobic methanotrophic archaea (e.g. ANME) contribute to methane consumption. Were ANME targeted or detected in this environment? or considered to play a role? It would be good to mention their existence at the very least.

3. Table 1: I do not believe this information is necessary in a table format. It should simply be listed in the methods.

4. Lines 286-289: It is stated that 6 of 7 methanogen orders were detected, it seems figure 6 should be referenced here, or this sentence moved to the next section.

5. Results title 3.3 “Methanogenic archaea and environmental OTUs”: This section discusses more than methanogenic archaea, so it seems odd that methanogenic archaea are specifically stated.

6. Line 324-325 and Fig. 6: Is site 6 really dominated by Methylacidiphilales? Based on the bar graph in Fig. 6 it looks like Methylococcales are a higher proportion. Also, due to the very low abundance of methyltrophs in the other sites, Figure 6 does not support the conclusion that type I and type II methylotrophs dominate the other sites, since the bars are too small.

7. Lines 387-390: It seems unlikely that differences in methane production rates between orders accounts for the differences in observed CH4 concentrations, and why high in site 6. I do not believe there is evidence for large differences in methane production under similar conditions. Substrate concentration and other environmental factors are largely responsible for methane production rates, not differences between individual species. It seems more likely that CH4 is highest in site 6, compared to the others, simply because there is far less sulfate. Methanogens at this site can outcompete SRBs for hydrogen produced by fermenting bacteria, where at all other sites sulfate is prevalent.

Also, many Methanobacteriales and Methanomicrobiales, which are the most abundant in this study, have been cultured.

Finally, what about Methanosarcinales? They were highly abundant in the sites with the most CH4 (4 and 6). Methanosarcinales are the most metabolically diverse methanogens and thrive in both freshwater and saltwater environments. In fact, in site 6 it appears Methanosarcinales were far more prevalent than Methanobacteriales. It is surprising that this was not addressed, but it should be.

8. Line 437-438: This is not a valid statement in the opinion of this reviewer. Aerobic methanotrophic bacteria are well characterized. The genes, enzymes, pathways, regulation etc. have all been studied. Also, it seems unlikely again that rates of methane oxidation (lines 435-437) is a valid argument for the observed differences.

6. PLOS authors have the option to publish the peer review history of their article (what does this mean?). If published, this will include your full peer review and any attached files.

Reviewer #1: No

Reviewer #2: No

---

## [Author Response · Author response to Decision Letter 0]

10 Oct 2020

We thank the Reviewers for their constructive comments. We have made the requested amendments and addressed the comments individually as detailed in the attached "Response to Reviewers" document.

---

## [Decision Letter · Decision Letter 1]

2 Nov 2020

Shifts in methanogenic archaea communities and methane dynamics along a subtropical estuarine land use gradient

PONE-D-20-09256R1

Dear Dr. Euler,

We’re pleased to inform you that your manuscript has been judged scientifically suitable for publication and will be formally accepted for publication once it meets all outstanding technical requirements.

Kind regards,

Ute Risse-Buhl, Ph.D.

Academic Editor

PLOS ONE

Additional Editor Comments (optional):

All comments have been carefully considered and are now addressed in the revised version. Therefore I recommend to accept the manuscript by Euler et al. for publication in PLOS ONE.

Reviewers' comments:

Reviewer's Responses to Questions

**Comments to the Author**

1. If the authors have adequately addressed your comments raised in a previous round of review and you feel that this manuscript is now acceptable for publication, you may indicate that here to bypass the “Comments to the Author” section, enter your conflict of interest statement in the “Confidential to Editor” section, and submit your "Accept" recommendation.

Reviewer #2: All comments have been addressed

2. Is the manuscript technically sound, and do the data support the conclusions?

Reviewer #2: Yes

3. Has the statistical analysis been performed appropriately and rigorously? 

Reviewer #2: Yes

4. Have the authors made all data underlying the findings in their manuscript fully available?

Reviewer #2: Yes

5. Is the manuscript presented in an intelligible fashion and written in standard English?

Reviewer #2: Yes

6. Review Comments to the Author

Reviewer #2: I have no additional comments to the authors. All comments have been addressed

I have no additional comments to the authors. All comments have been addressed

7. PLOS authors have the option to publish the peer review history of their article (what does this mean?). If published, this will include your full peer review and any attached files.

Reviewer #2: No

---

## [Editor Report · Acceptance letter]

11 Nov 2020

PONE-D-20-09256R1 

Shifts in methanogenic archaea communities and methane dynamics along a subtropical estuarine land use gradient 

Dear Dr. Euler:

I'm pleased to inform you that your manuscript has been deemed suitable for publication in PLOS ONE. Congratulations! Your manuscript is now with our production department. 

Kind regards, 

on behalf of

Dr. Ute Risse-Buhl 

Academic Editor

PLOS ONE